# Systematic review and meta-analysis of developmental assets scales: A study protocol for psychometric properties

**Mojtaba Habibi Asgarabad**[1]*, **Pardis Salehi Yegaei**[2], **Elizabeth Trejos-Castillo**[3,4], **Nazanin Seyed Yaghoubi Pour**[5], **Nora Wiium**[6]

1 Department of Psychology, Norwegian University of Science and Technology, Trondheim, Norway,
2 Health Promotion Research Center, Iran University of Medical Science, Tehran, Iran, 3 Department of Health Psychology, School of Behavioral Sciences and Mental Health (Tehran Psychiatric Institute), Iran University of Medical Sciences, Tehran, Iran, 4 Human Development & Family Sciences, Texas Tech University, Lubbock, Texas, United States of America, 5 Department of Psychology, Faculty of Education and Psychology, University of Tabriz, Tabriz, Iran, 6 Department of Psychosocial Science, Faculty of Psychology, University of Bergen, Bergen, Norway

* Mojtaba.h.asgarabad@ntnu.no

**Data Availability Statement:** Deidentified research data will be made publicly available when the study is completed and published.

## Abstract

### Background and aims

Application of developmental assets, one of existing Positive Youth Development (PYD) frameworks, has gained momentum in research, policy formulations, and interventions, necessitating the introduction of the most efficient scales for this framework. The present study protocol aims to conduct a systematic review and meta-analysis of developmental assets scales to document the underlying logic, objectives, and methodologies earmarked for the identification, selection, and critical evaluation of these scales.

### Methods and materials

In accordance with the Preferred Reporting Items for Systematic Review and Meta-Analysis Protocols (PRISMA-P), the intended search will encompass databases of PubMed, Scopus, Web of Knowledge, and PsycINFO, spanning from the inception of 1988 to 1st of April 2024. The review will include articles published published in English language focusing on individuals aged 10 to 29 years and reporting at least one type of reliability or validity of developmental assets scales. The review process will be in compliance with the COnsensus-based Standards for the selection of health Measurement INstruments (COSMIN), and the overall quality of evidence will be determined using the Grading of Recommendations, Assessment, Development and Evaluations (GRADE) guidelines.

### Discussion

This comprehensive assessment aims to identify potential biases in prior research and offer guidance to scholars regarding the optimal scales for developmental assets in terms of validity, reliability, responsiveness, and interpretability The evidence-based appraisal of the scales strengths and limitations is imperative in shaping future research, enhancing their

**Funding:** Thanks go to the Norwegian University of Science and Technology (NTNU) for providing financial support for the publication of this article as an open-access article. The funders played no role in the design of the study, the collection of data, the analysis, the decision to publish, or the preparation of the manuscript.

**Competing interests:** The authors have no commercial or financial relationships that could be construed as a potential conflict of interest.

methodological rigor, and proposing refinements to existing instruments for developmental assets.

## 1. Introduction

Positive youth development (PYD) is a strength-based developmental perspective that puts emphasis on meaningful and constructive involvement of young individuals in their communities, educational institutions, social circles, and families with the aim of empowering and enabling them to achieve their maximum capabilities [1]. PYD concentrates on boosting young people's strengths, establishing supportive contexts, and promoting reciprocal and constructive youth ↔ context interactions [2, 3].

PYD has gained ground in organizations, community-based services, and youth-serving programs providing opportunities for young people to foster their competencies [4]. Thus, an array of frameworks over the past decades has been conceptualized and designed to define and capture PYD. The Search Institute in collaboration with Benson et al. [5] proposed the conceptual framework of PYD that includes 40 developmental assets, comprising 20 individual assets and contextual assets (e.g., home, school, community). To assess this model, Benson et al. [6] developed the 58-item developmental assets profile (DAP) which includes two asset types: a) the "asset category" perspective that organizes items into measures representing eight internal and external developmental asset categories (Table 1); and b) "asset context" perspective that regroup items based on how young individuals experience these assets in various ecological contexts (i.e., personal, family, school, social, and community). DAP items are scored using a scale ranging from "*not at all/rarely*" to "*extremely/almost always.*" In terms of psychometric properties, the DAP is found to possesses invariance over time [7], and has acceptable to good reliability and validity in both individual projects and group aggregates [6]. For instance, higher scores in the developmental assets have been linked to adolescent achievement [8], avoidance of risky behaviors [9], and improvement of pro-social behavior, resiliency, and leadership [7, 10].

Another scale, Youth Asset Survey (YAS; [11]), was designed to assess the associations of developmental assets with risky behaviors in a prospective study on adolescents and their parents. This 37-item survey encompasses eight subscales corresponding to eight developmental assets of family communication, peer role models, general future aspirations, responsible choices, community involvement, non-parental role models, use of time on groups/sports, and use of time on religion, along with two one-item subscales of cultural respect and good health practices (exercise/nutrition). Although the construct validity and internal consistency of YAS has been supported, subscales of family communication and future aspirations have reported low alpha coefficients ($< .68$). Additionally, the number of items was limited for two additional assets; cultural respect and good health practices, including exercise and nutrition [11]. In an attempt to modify this scale, Oman, Lensch [12] conducted a longitudinal cohort study and provided an improved 68-item scale, Youth Asset Survey -Revised (YAS-R). The YAS-R appraises seven additional developmental assets, namely, religiosity, school connectedness, relationship with father, relationship with mother, general self-confidence, parental monitoring, and educational aspirations for the future.

Taken together, as PYD programs and frameworks grow in popularity, there is a compelling need to globally and culturally adapt appropriate and relevant measures of developmental assets that are psychometrically sound across various contexts [13]. The extensive utilization of these scales warrants a systematic review to examine the caliber of their

**Table 1. Overview of sample developmental assets scales and their characteristics.**

| Scale | Target population (Administration method) | Number of items (Scoring type) | Subscales | Alpha |
|---|---|---|---|---|
| Developmental Assets Profile (DAP) [6] | Adolescents aged 8–11 & 12–18 (Self-report) | 58 (Likert scale) | 1. Support<br>2. Empowerment<br>3. Boundaries & expectations<br>4. Constructive use of time<br>5. Commitment to learning<br>6. Positive values<br>7. Social competencies<br>8. Positive identity | .80<br>.74<br>.84<br>.36<br>.83<br>.85<br>.79<br>.79 |
| Youth Asset Survey (YAS) [11] | Adolescents (Self-report) | 37 (Likert scale) | 1. Family communication<br>2. Peer role models<br>3. General future aspirations<br>4. Responsible choices<br>5. Community involvement<br>6. Non-parental role models<br>7. Use of time on groups/sports<br>8. Use of time on religion<br>9. Cultural respect<br>10. Good health practices (exercise/nutrition) | .61<br>.81<br>.67<br>.69<br>.78<br>.74<br>.71<br>.71<br>- (One item)<br>- (One item)<br>[11] |
| Youth Asset Survey -Revised (YAS-R) [12] | Adolescents (Self-report) | 68 (Likert scale) | 1. Family communication<br>2. Peer role models<br>3. General future aspirations<br>4. Responsible choices<br>5. Community involvement<br>6. Non-parental role models<br>7. Use of time on groups/sports<br>8. Use of time on religion<br>9. Cultural respect<br>10. Good health practices (exercise/nutrition)<br>11. Religiosity<br>12. School connectedness<br>13. Relationship with father<br>14. Relationship with mother<br>15. General self-confidence<br>16. Parental monitoring<br>17. Educational aspirations for the future | .77<br>.83<br>.77<br>.76<br>.89<br>.82<br>.79<br>.76<br>.76<br>.84<br>.91<br>.73<br>.91<br>.89<br>.74<br>.90<br>.72 [12] |

psychometric properties, describe their plausible psychometric shortcomings and strengths, and identify the best measures for researchers. Despite the availability of numerous developmental assets scales, the comparison of psychometric characteristics of the most employed scales have been mostly understudied. Therefore, there is a pressing need for a summary of available developmental assets scales and their psychometric robustness to serve as a guide for choosing the right measurement tool when conducting investigations and implementing youth programs.

Furthermore, a growing body of evidence is concerned with the prevalence of publication bias in the context of systematic reviews [e.g., 14] suggesting that the failure to publish complete studies might pose a challenge for systematic reviews. For instance, Silagy, Middleton [15]' study underscored the potential bias towards favoring positive findings in published systematic reviews. Given these observations, it becomes paramount that a predefined protocol is established before the review, articulating whether the review outcomes align with the original study plan, and enhancing the transparency in both the execution and eventual reporting of systematic review [16]. Thus, a review protocol that is subjected to peer review contributes to

preventing ad hoc decisions in the review process, as well as reducing publication bias and selective reporting [17, 18].

The proposed systematic review is the first-of-its-kind study that will fill the existing gap on the characteristics and psychometric properties of measures of developmental assets. In particular, our objectives are to: 1) prepare a comprehensive list of available tools developed for developmental assets, 2) summarize the characteristics of these tools/questionnaires (e.g., number of components/items, assessment method, language, and scoring type), 3) identify the most commonly used psychometric indexes for the evaluation of these tools/measures (e.g., reliability, validity, measurement error, responsiveness, and interpretability), 4) appraise the extent to which the measurement properties of these tools/questionnaires possess the methodological quality in accordance with the COnsensus-based Standards for the selection of health Measurement INstruments (COSMIN) criteria [19], and 5) compare the quality of the measurement properties related to these tools based on the results of COSMIN (if applicable).

## 2. Methods

Reporting the proposed review aligns with the guidelines of the Preferred Reporting Items for Systematic Review and Meta-Analyses (PRISMA) [20]. See S1 Table for a completed PRISMA-P-checklist.

### 2.1. Eligibility criteria

The three main eligibility criteria for papers containing developmental assets scales are as follows: a) being published in English, b) targeting young people aged 10–29 in line with Catalano, Skinner [21] suggestion, and c) being published after 1998 (when developmental assets were first conceptualized by Benson, Leffert [5]). Besides these primary criteria, studies must have followed at least one of the following aims: 1) reporting quantitative information on the appropriateness or acceptability of the measurement tools for developmental assets, 2) providing at least one of the reliability indexes of these tools, 3) containing information on the validity of these tools, or 4) using these tools to evaluate risk factors (i.e., predictor variable) and/or study outcomes. Papers with different methodological designs, such as cohort studies, randomized/non-randomized controlled trials, cross-sectional, post-intervention, and case-control studies, as well as grey literature will be included if they meet eligibility criteria. Studies will be excluded if they: 1) contain no empirical evidence (i.e., theoretical framework discussions and editorials) and 2) are literature reviews.

### 2.2. Information sources and strategy for search

A quick literature review based on Medical Subject Heading (MeSH) [22] will be used to identify keywords in two domains, namely, "developmental assets," and "tools/ questionnaire" [the extended keywords are presented in syntax search in Table 2]. A preliminary search strategy will be developed with the aid of a senior librarian. A sample search strategy for PubMed is presented in Table 2 (This syntax is subject to vary based on the final search strategy). This search strategy has been adjusted according to the second version of the COSMIN initiative's search filter [23]. Each domain will be searched individually to launch pilot searches. Following that, a comprehensive search will be conducted by combining all domains to ensure that an appropriate search strategy is implemented. Subsequently, the databases Scopus, PubMed, PsycINFO, and Web of Science will be searched starting on the establishment date of the developmental assets framework (1998) through the 1st of April 2024 (This date is subject to vary based on the final date of coverage). To include additional studies and explore references further, we will check the studies' references and conduct a broader search across scientific

**Table 2. Search strategy for PubMed database.**

| | Search Strategy | Number of records |
|---|---|---|
| #1 | Positive Youth Development[Title/Abstract] | 908 |
| #2 | (instrumentation[sh] OR methods[sh] OR "Validation Studies" OR "psychometrics" OR psychometr*[tiab] OR "outcome assessment"[tiab] OR "outcome measure*" OR "discriminant analysis" OR reliab*[tiab] OR unreliab*[tiab] OR valid*[tiab] OR "internal consistency"[tiab] OR (cronbach*[tiab] AND (alpha[tiab] OR alphas[tiab])) OR (item[tiab] AND (correlation*[tiab] OR selection*[tiab] OR agreement OR test-retest [tiab] OR (test[tiab] AND retest[tiab]) OR (reliab*[tiab] AND (test[tiab] OR retest[tiab])) OR stability[tiab] OR interrater[tiab] OR inter-rater [tiab] OR intrarater[tiab] OR intra-rater[tiab] OR intertester[tiab] OR intratester[tiab] OR intratester[tiab] OR intra-observer[tiab] OR kappa [tiab] OR kappa's[tiab] OR kappas[tiab] AND (measure OR measures OR (intraclass[tiab] AND correlation*[tiab]) OR "factor analysis"[tiab] OR "factor analyses"[tiab] OR "factor structure"[tiab] OR "factor structures"[tiab] OR dimension*[tiab] OR subscale*[tiab] OR (multitrait[tiab] AND scaling[tiab] AND (analysis[tiab] OR analyses[tiab])) OR "item discriminant"[tiab] OR "interscale correlation*"[tiab] OR error[tiab] OR errors[tiab] OR (uncertainty[tiab] AND (measurement[tiab] OR measuring[tiab])) OR "standard error of measurement"[tiab] OR "Item response model"[tiab] OR IRT[tiab] OR Rasch[tiab] OR "Differential item functioning"[tiab] OR DIF[tiab] OR "computer adaptive testing"[tiab] OR "item bank"[tiab] OR "cross-cultural equivalence"[tiab]) | 610116 |
| #3 | ((((((((((((((((((((((((((((((((((((((((((((((((((((((((((Surveys and Questionnaires[MeSH Terms]) OR ("Surveys and Questionnaires"[Title/Abstract])) OR (Survey Methods[MeSH Terms])) OR ("Survey Methods"[Title/Abstract])) OR (Questionnaires and Surveys[MeSH Terms])) OR ("Questionnaires and Surveys"[Title/Abstract])) OR (Methods, Survey[MeSH Terms])) OR ("Methods, Survey"[Title/Abstract])) OR (Survey Method[MeSH Terms])) OR ("Survey Method"[Title/Abstract])) OR (Methodology, Survey[MeSH Terms])) OR ("Methodology, Survey"[Title/Abstract])) OR (Survey Methodology[MeSH Terms])) OR ("Survey Methodology"[Title/Abstract])) OR (Surveys[MeSH Terms])) OR (Surveys[Title/Abstract])) OR (Survey[MeSH Terms])) OR (Survey[Title/Abstract])) OR (Questionnaire Design[MeSH Terms])) OR ("Questionnaire Design"[Title/Abstract])) OR (Design, Questionnaire[MeSH Terms])) OR ("Design, Questionnaire"[Title/Abstract])) OR (Designs, Questionnaire[MeSH Terms])) OR ("Designs, Questionnaire"[Title/Abstract])) OR (Questionnaire Designs[MeSH Terms])) OR ("Questionnaire Designs"[Title/Abstract])) OR (Baseline Survey[MeSH Terms])) OR ("Baseline Survey"[Title/Abstract])) OR (Baseline Surveys[MeSH Terms])) OR ("Baseline Surveys"[Title/Abstract])) OR (Survey, Baseline[MeSH Terms])) OR ("Survey, Baseline"[Title/Abstract])) OR (Surveys, Baseline[MeSH Terms])) OR ("Surveys, Baseline"[Title/Abstract])) OR (Respondents [MeSH Terms])) OR (Respondents[Title/Abstract])) OR (Respondent[MeSH Terms])) OR (Respondent[Title/Abstract])) OR (Questionnaires [MeSH Terms])) OR (Questionnaires[Title/Abstract])) OR (Questionnaire[MeSH Terms])) OR (Questionnaire[Title/Abstract])) OR ((((((((Self Report[MeSH Terms]) OR ("Self Report"[Title/Abstract])) OR (Report, Self[MeSH Terms])) OR ("Report, Self"[Title/Abstract])) OR (Reports, Self[MeSH Terms])) OR ("Reports, Self"[Title/Abstract])) OR (Self Reports[MeSH Terms])) OR ("Self Reports"[Title/Abstract])) OR (index)) OR (indices)) OR ("self-report")) OR ("self-report measures")) OR ("assessment tools")) OR ("measurement scale")) (instrument)) OR ("measurement instrument")) OR (scale)) OR (measure)) OR (tool) | 5,887,880 |
| #4 | #1 AND #2 AND 3 | 105 |

Filters applied: Full text, Humans, English, from 1998/1/1–2024/1/1

journals covering related fields. Furthermore, we will contact specialists in the field to obtain unpublished or under-review papers, where possible. Ultimately, we will search gray literature through the Healthcare Management Information Consortium (HMIC) and the European Association for Grey Literature Exploitation (EAGLE).

## 2.3. Data screening

The Rayyan QCRI online software [24] will be utilized to organizing references, titles, and abstracts of the papers and identifying duplicates. The titles and abstracts will be examined in the screening stage; papers that are not compatible with the proposed study's purpose will be excluded. In the eligibility phases, the entire articles will be reviewed to determine if they fit the inclusion criteria to be considered for the meta-analysis. An independent reviewer will assess the manuscripts (PSY) and a senior researcher (MHA) will review the results. The details of the screening procedure are displayed in the flowchart of the PRISMA extension for systematic reviews (Fig 1).

## 2.4. Data extraction

A data extraction form will be designed in Microsoft Excel 2016 (a sample form is presented in Table 3). In the following step, data from three papers will be extracted to identify and modify possible flaws and deficiencies in the data extraction form. Two expert researchers (PSY and

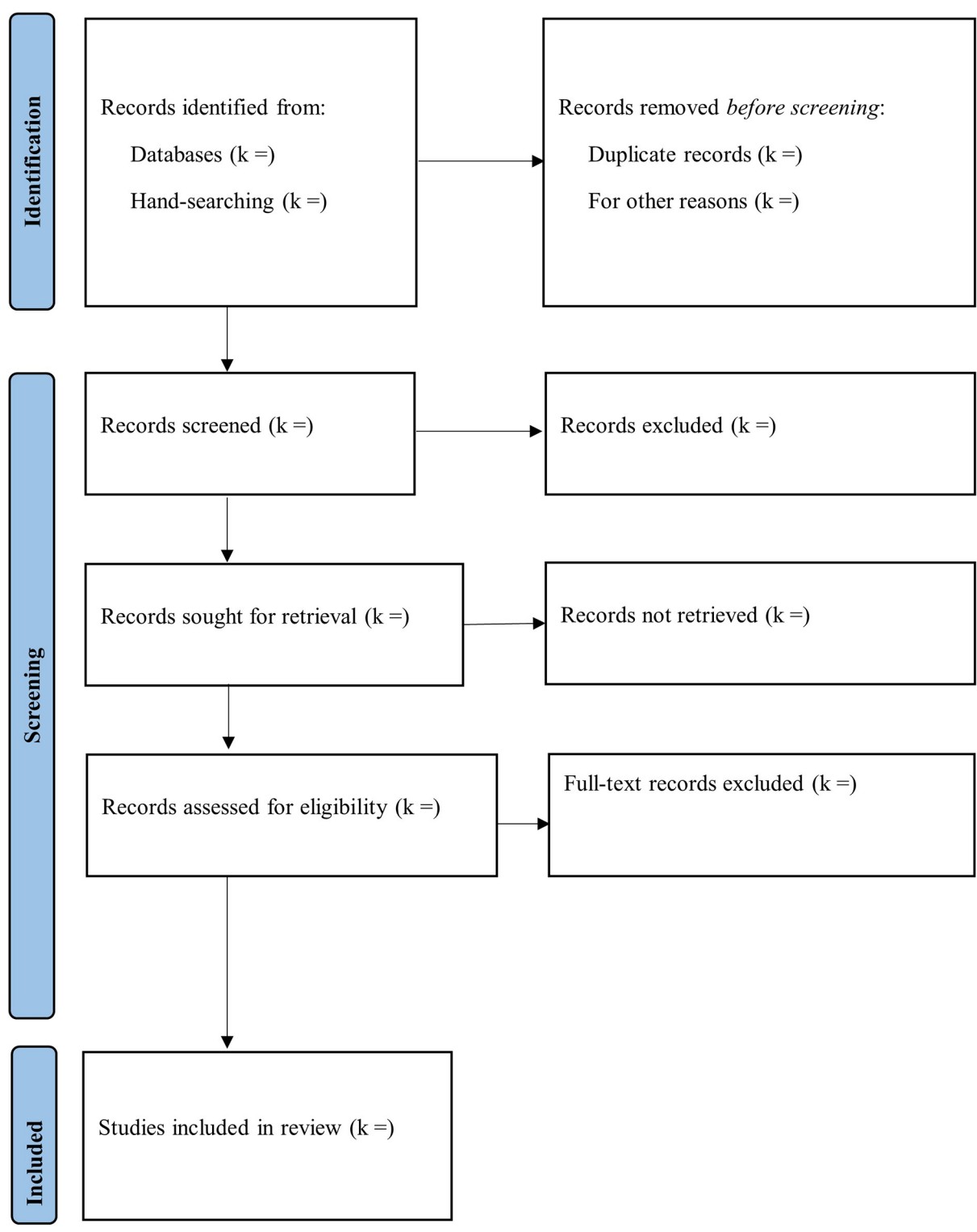

**Fig 1. The PRISMA flowchart of search and screening process.**

**Table 3. Preliminary data extraction form.**

| Data | Included studies | | | | |
|---|---|---|---|---|---|
| | 1 | 2 | 3 | 4 | 5 |
| Author(s) (Published year) | | | | | |
| Country | | | | | |
| Language | | | | | |
| Participants (N; age; gender) | | | | | |
| Country income level (family income level) | | | | | |
| Minority/majority | | | | | |
| Study setting | | | | | |
| Measurement tool title | | | | | |
| Measurement tool development/adaptation | | | | | |
| Item formulation type | | | | | |
| Assessment method (scoring type) | | | | | |
| Number of subscales (number of items) | | | | | |
| Internal consistency | | | | | |
| Measurement error | | | | | |
| Content validity | | | | | |
| Face validity | | | | | |
| Construct validity | | | | | |
| Structural validity | | | | | |
| Cross-cultural validity | | | | | |
| Criterion validity | | | | | |
| known-group validity | | | | | |
| Longitudinal validity | | | | | |
| Responsiveness | | | | | |
| Interpretability | | | | | |

*Notes.* N = number.

MHA) will extract the information from chosen papers individually, and, in case of any ambiguity, a senior researcher will further evaluate the extraction process (NW). In cases where data in the papers are missing, we will contact the authors and ask for original data via Email. The final form of extracted data will contain details on authors, year of publication, country, language, sample size, participant's age and gender, classification of country and family by income level (ranging from low to high income), minority group, study setting (e.g., family/home, community, or school), measurement tool title, measurement tool development (with objectives) or adaptation, how initial questions were generated (e.g., theory- and literature review-derived, expert panel, focus group discussion, combining previous measurement tools), assessment method (self/proxy/teen/teacher-report questionnaire, interview, observation), scoring type (multiple-choice, Likert, etc.), number of subscales and items, reliability (i.e., internal consistency and measurement error), validity (i.e., face, content, construct, structural, cross-cultural, criterion, known-group, and longitudinal validity), responsiveness, and interpretability.

## 2.5. Outcomes

Expected outcomes of this study will include offering an exhaustive and clear description of the measurement tools accessible for developmental assets and discovering possible

**Table 4. Defining the domains, sub-domains, and aspects of measurement properties using COSMIN.**

| Domain | Measurement property | Aspects | Definition |
|---|---|---|---|
| Reliability | | | Presenting consistent scores in similar conditions |
| | 1. Internal consistency | | The extent the items of a measurement tool are inter-correlated |
| | 2. Measurement error | | The dissimilarity between the unobserved true value of a variable and its observed value |
| Validity | | | The extent to which a measurement tool assesses the construct it is intended to assess |
| | 1. Content validity | | A measurement tool's ability to adequately capture all aspects of the intended latent construct |
| | | 1.1 Face validity | The extent to which a measurement tool's items truly reflect the construct they were developed to measure |
| | 2. Construct validity/ Hypotheses testing | | The level that the measurement tool's scores are compatible with hypotheses or theoretical model |
| | | 2.1 Structural validity | How well a measurement tool is able to reflect the dimensionality of the latent construct it was developed to measure |
| | | 2.2 Cross-cultural validity/ measurement invariance | How well a measurement tool can function similar to the original tool in various cultural groups |
| | 3. Criterion validity | | The extent to which a measurement tool's scores are correlated with a "gold standard" measure |
| | 4. known-group validity | | How well the PROM is able to distinguish the subgroups of a sample |
| Responsiveness | | | The capability of a measurement tool to discover the change in the scores of the aimed construct over time |
| | 1. Longitudinal validity | | How well a measurement tool is able to discover the "expected changes" in the scores of the aimed construct over time |
| Interpretability | | | How well the qualitative meaning (i.e., everyday understanding or clinical meaning) can be assigned to a measurement tool's scores |

*Notes*. The Table is derived from Mokkink, Terwee [26] and Mokkink, Terwee [27]. Interpretability is not one of the measurement properties, but a critical characteristic of a measure.

shortcomings and strengths of these measurement tools. Additional outcomes may include: a) aiding researchers to select appropriate measurement tools in future investigations, and b) assisting researchers in selecting appropriate tool by assessing the utility and adaptability of the chosen instrument in their region and cultural context.

## 2.6. Potential biases in single studies

To facilitate the evaluation of risk of bias in every research study, we will apply the COSMIN criteria for systematic reviews of PROMs [19, 25]. The 116-item checklist of COSMIN's Risk of Bias consists of ten criteria, as detailed in Table 4, including validity (e.g., content, structural, and criterion validity), reliability (e.g., stability and measurement error), responsiveness, and interpretability. The assessment uses a four-point rating scale: "*very good*", "*adequate*", "*doubtful*", and "*inadequate*". The quality of measurement properties will be graded as *sufficient* (+), *insufficient* (-), or *indeterminate* (?) based on the COSMIN criteria for good measurement properties.

## 2.7. Data synthesis and meta-analysis

Prior to data synthesis and if feasible, data pooling will be conducted. A standard psychometric meta-analysis approach, posited by Hunter and Schmidt [28, 29] will be performed based on the psychometrics principles. This approach suggests that measurement errors (caused by

unreliable measures), errors in sampling process, and range limitations are some of the sources that cause artifact variability and account for a large portion of the observed variation in the relationship between two variables in original studies. Consequently, it is crucial to conduct meta-analyses to identify potential moderating factors influencing these relationships and address artifact variability across studies. By identifying these factors, researchers can control artifact variability through either selecting appropriate study designs or subtracting artifact validity from the overall observed variability.

Meta-analysis will be based on Fisher's Z transformed partial correlation coefficient, known to have the lowest root mean square error and bias [30] in comparison with the partial correlation coefficients of meta-analysis [31]. The standardized effect size will be Fisher's Z that ranges from $-\infty$ to $+\infty$ and the standards used to interpret them are similar to those used for a correlation coefficient. If intraclass, Pearson, or Spearman correlations are provided, we will apply the Fisher's variance stabilizing transformation [32, 33] to convert them into Fisher's Z scores. If the coefficients of unstandardized beta and $F$-ratios are provided, we will primarily convert them to $r$ and afterwards to Fisher's Z score [32, 33]. If only $p$ values are provided, we will convert them to Z-score, $r$, and Fisher's Z, respectively [33]. We will extract the overall effect size for each psychometric property and the effect sizes for each follow-up interval from studies that include follow-up assessments.

Data analysis will be carried out utilizing Comprehensive Meta-Analysis v.3 software [34]. With the presumption that heterogeneity is probable and that the mean of the effect size is not stable across studies, random-effects models will be applied. To assess heterogeneity, the Cochrane's Q test (the presence of heterogeneity) and the $I^2$ statistic (diversity in heterogeneity effect estimates) will be utilized [35]. Based on the standard interpretation [36], $I^2$ statistic will be defined as "not important" (0–40%), "moderate" (30–60%), "substantial" (50–90%), and "considerable" (75–100%). Furthermore, where appropriate, funnel plots will be provided to identify reporting bias and the effects size of small studies [36]. To ensure that the meta-analysis's findings are robust in case of considerable heterogeneity, a sensitivity analysis will be conducted.

We will create a qualitative abstract based on studies' outcomes concerning the measurement properties of each measurement tool when it is not feasible to pool the data. If there are discrepancies between the findings across various studies, possible explanations will be provided. If a consistent pattern appears, we will consolidate the results for each subgroup with consistent findings. In case that no clear justifications or discernible patterns emerge, the majority of the findings will be employed to assess the results.

## 2.8. Quality assessment of individual studies

As described in Table 5, the quality of results of every study will be evaluated via the COSMIN criteria for good measurement properties. These properties will be rated as "*insufficient*, (-)" "*indeterminate*, (?)" or "*sufficient* (+)". The total score of a property will be equal to the lowest score it obtains, and its interpretation will be on the basis of the COSMIN criteria: 50% (high quality), 30–50% (moderate quality), and less than 30% (low quality) [19, 37, 38]. Four domains of reliability, validity, responsiveness, and interpretability will be included in the quality assessment taxonomy. This study will assess the quality of Exploratory Factor Analysis (EFA) using the guidelines outlined by Terwee, Bot [39]. According to these guidelines, in the absence of a theoretical or empirically emerged structural model, EFA is preferable. In contrast, when a model has already been theoretically proposed and/or has empirically emerged in the literature, should Confirmatory Factor Analysis (CFA) be tested [40, 41]. The results of EFA's quality assessment will be interpreted as follows: (+) the chosen factors can explain at

**Table 5. COSMIN criteria for good measurement properties.**

| Measurement property | Rating | Criteria |
|---|---|---|
| Structural validity | + | 1. CTT: One of the following criteria in CFA should be met:<br>• CFI/ TLI/ comparable measure >.95<br>• SRMR < .08<br>• RMSEA < .06<br>2. IRT/Rasch: All of the following criteria should be met:<br>• No violation of unidimensionality: CFI/ TLI/ comparable measure >.95 OR SRMR < .08 OR RMSEA < .06<br>• No violation of local independence: residual correlations among the items after controlling for the dominant factor < .20 OR Q3's < .37<br>• No violation of monotonicity: adequate looking graphs/ item scalability >.30 and adequate model fit: IRT: $\chi^2$>.01; Rasch: infit and outfit mean squares $\geq$.5 and $\leq$1.5 OR Z standardized values > −2 and <2 |
| | ? | 1. CTT: Not reporting all information for "+"<br>2. IRT/Rasch: Not reporting model fit |
| | - | Not meeting criteria for "+" |
| Internal consistency | + | At least low evidence for sufficient structural validity (assessed via GRADE) and $\alpha$(s)$\geq$0.70 for each unidimensional scale or subscale |
| | ? | Not meeting criteria for low evidence or higher |
| | - | At least low evidence for sufficient structural validity and Cronbach's alpha(s) < .70 for each unidimensional scale or subscale |
| Reliability | + | ICC/ weighted Kappa $\geq$.70 |
| | ? | ICC/ weighted Kappa not reported |
| | - | ICC/ weighted Kappa < .70 |
| Measurement error | + | SDC/ LoA < MIC |
| | ? | MIC is not defined |
| | - | SDC/ LoA >MIC |
| Hypotheses testing/construct validity | + | Most evidence in studies (75% or more) are in line with the hypotheses |
| | ? | Hypotheses are not defined (by review team) |
| | - | Evidence are not in line with the hypotheses |
| Cross-cultural validity/measurement invariance | + | No important difference between group factors (e.g., age) in multiple group factor analysis OR no important DIF for group factors (McFadden's $R^2$ < .02) |
| | ? | The absence of multiple group factor analysis or DIF analysis |
| | - | Important differences between group factors or DIF |
| Criterion validity | + | Correlation with gold standard/ AUC $\geq$.70 |
| | ? | Not reporting all information for sufficient rating |
| | - | Correlation with gold standard/ AUC < .70 |
| Responsiveness | + | Results are in line with the hypothesis or AUC $\geq$.70 |
| | ? | No hypotheses was defined (by review team) |
| | - | Results are not in line with the hypotheses or AUC < .70 |

*Notes.* The Table is derived from Prinsen, Mokkink [37] and Mokkink, Prinsen [42]. "+" = Sufficient, "?" = Indeterminate, "-" = Insufficient, CTT = Classical test theory, CFA = Confirmatory factor analysis, IRT = Item response theory, CFI = Comparative fit index, TLI = Tucker-Lewis index, RMSEA = Root mean square error of approximation, SRMR = Standardized root mean residuals, $\alpha$ = Cronbach's alpha, GRADE = Grading of Recommendations, Assessment, Development and Evaluations, ICC = Intra-class correlation coefficient, SDS = Smallest detectable change, LoA = Limits of agreement, MIC = Minimal important change, DIF = Differential item functioning, AUC = Area under curve.

least 50% of the variance OR they can explain less than 50% of the variance but a justification for this selection is proposed by authors; (?) the vague or incomplete information (e.g., failure to mention the explained variance) prevents scoring the EFA's quality; and (-) criteria for a "plus" rating was not met [39].

## 2.9. Confidence in consolidated findings

The Grading of Recommendations, Assessment, Development, and Evaluation (GRADE) working group approach [43] will be used to test the credibility of each piece of research posterior to provide a summary of general ratings on each psychometric property. The quality of evidence will be examined using five categories: risk of bias, publication bias, imprecision, inconsistency, and indirectness. Two researchers (PSY and MHA) will independently appraise the overall quality of the summarized findings. In the event of any disagreement, a third researcher (ETC) will further review the findings. The quality of outcomes will be classified as *high* (indicating a high degree of confidence in the measurement property's estimate being close to the true value), *moderate* (indicating a reasonable belief that the true estimate of the measurement property is likely close to the estimated value), *low* (suggesting a substantial potential for a significant difference between the true estimate and the estimated property), or *very low* (indicating a high likelihood of a substantial deviation between the actual measurement property and its estimated value).

## 2.10. Recommendations for instrument choice

The assessment of the psychometric soundness of the measurement tools for developmental assets, along with providing recommendations for their forthcoming applications will be carried out in accordance with a composite of general ratings for each psychometric property and the grading outcomes, as outlined by Prinsen, Mokkink [37]. The outcomes derived from all included studies pertaining to each measurement property will be stratified into three distinct recommendation classifications, as articulated by Mokkink, Prinsen [42] and Mokkink, De Vet [19]: (a) the developmental assets scale has the potential to be introduced as the most suitable instrument for evaluating its intended theoretical model; (b) the scale may be tentatively endorsed, though further investigations into its validation are imperative; and (c) the scale should not be endorsed. The rationale underlying the assignment of each measurement tool to the aforementioned categories will be further explained. Subsequently, prospective trajectories for research will be delineated, where pertinent.

## 2.11. Ethics and dissemination

As this project did not recruit participants, it is not a prerequisite to obtain ethical approval. The findings of the intended review will be reported in a peer-reviewed journal.

## 3. Discussion

The present study protocol aims to systematically review the underlying logic and aims of developmental assets scales, and to specify the methods that will be employed to critically evaluate primary studies on these scales. The main objective of the proposed systematic review and meta-analysis is to compile, summarize, and critically evaluate the psychometric properties (i.e., validity, reliability, measurement error, interpretability, and responsiveness) of measurement tools including but not limited to: a) the 58-item developmental assets profile (DAP) [6] derived from the developmental assets model proposed by Benson, Leffert [5], Youth Asset Survey (YAS; [11]), and Youth Asset Survey -Revised (YAS-R) [12].

Given that the PYD frameworks including developmental assets have gained traction in various areas, such as youth research, policy formulations [44], school-based organization programs [45], adolescents behavior interventions [46], community-based health services [47], socializing systems [48], and cross-sectoral interventions in education [49], it is crucial to identify the most efficient and preferred measures for developmental assets. To the best of our knowledge, no

systematic review of the psychometric properties of developmental assets scales has been carried out to date. Thus the proposed systematic review will be the first. In general, we strive to incorporate an in-depth summary and collection of the most used measurement tools in the developmental assets framework, the purpose of their development, their characteristics, and psychometric indexes. In addition, we aim to delve into the methodological quality of their measurement properties based on COSMIN criteria. Consequently, by discerning the strength of those measurement tools, we wish to assist researchers, clinicians, and community-health specialists in making informed choices and select more optimal measures aligned with their desired objectives. Besides, through the detection of potential shortcomings, researchers may underscore the need for enhancing current measures. Furthermore, considering the rigorous quality assessments of the aforementioned scales, the level of validity, reliability, feasibility, productivity, as well as potential risk of bias of earlier published studies will become evident.

The present study protocol has several implications. First, the rigor and reliability of a systematic review on the developmental assets scales are contingent upon the meticulous pre-planning that preemptively identifies potential challenges [18]. In the process of publication, a documented protocol undergoes the rigorous peer review that scrutinize it and guarantees its appropriateness prior to its publication [16]. The second factor that ensures the trustworthiness of a systematic review is the thorough documentation of the methodology used in the review process prior to commencing the review. This documentation permits other researchers to compare the protocol with the finalized review, thus identifying instances of selective reporting or deviations and assess the proposed methodologies' validity [17]. Third, a study protocol prevents making arbitrary decisions regarding inclusion of studies and data extraction, and mitigates publication bias in favor of only reporting "positive" findings (or those findings in line with authors' hypotheses) [50]. Finally, a properly conducted research protocol allows for the potential replication of (revised) review methods. Given that the current protocol does not include measurement tools and studies in other languages, local researchers will receive a published protocol, which they can adapt for use in their context and language, with some modifications.

The present study protocol is not without potential limitations. The first drawback is the exclusion of locally standardized/developed developmental assets scales that are not accessible in English. As a result, we will possibly miss significant related research published in other languages. Second, given that existing studies using developmental assets scales have been carried out in different locations and periods, the accuracy of the assessment may have been inadvertently affected by the external/environmental factors. Third, since research incorporates a wide range of study methods and samples, the assessment of statistical heterogeneity is frequently overlooked or reported insufficiently. Hence, data synthesis and meta-analysis may be impacted by high rates of heterogeneity.

## Supporting information

**S1 Table. PRISMA-P (Preferred Reporting Items for Systematic review and Meta-Analysis Protocols) 2015 checklist: Recommended items to address in a systematic review protocol.** (DOC)

## Acknowledgments

The authors gratefully thank those who kindly participated in this research.

## Author Contributions

**Conceptualization:** Mojtaba Habibi Asgarabad.

**Investigation:** Mojtaba Habibi Asgarabad, Pardis Salehi Yegaei, Elizabeth Trejos-Castillo, Nora Wiium.

**Methodology:** Mojtaba Habibi Asgarabad, Pardis Salehi Yegaei.

**Project administration:** Mojtaba Habibi Asgarabad, Pardis Salehi Yegaei.

**Supervision:** Mojtaba Habibi Asgarabad.

**Validation:** Mojtaba Habibi Asgarabad, Pardis Salehi Yegaei, Nazanin Seyed Yaghoubi Pour.

**Writing – original draft:** Mojtaba Habibi Asgarabad, Pardis Salehi Yegaei, Nazanin Seyed Yaghoubi Pour.

**Writing – review & editing:** Mojtaba Habibi Asgarabad, Pardis Salehi Yegaei, Elizabeth Trejos-Castillo, Nazanin Seyed Yaghoubi Pour, Nora Wiium.

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
