## [Decision Letter · Decision Letter 0]

13 Feb 2024

PONE-D-23-40747Systematic Review and Meta-Analysis of Developmental Assets Scales: A Study Protocol for Psychometric PropertiesPLOS ONE

Dear Dr. Habibi Asgarabad,

Thank you for submitting your manuscript to PLOS ONE. After careful consideration, we feel that it has merit but does not fully meet PLOS ONE’s publication criteria as it currently stands. Therefore, we invite you to submit a revised version of the manuscript that addresses the points raised during the review process.

**Dear Authors**

**We received the reviewer's comment. Please respond to the comment point by point.**

**Cheers**

We look forward to receiving your revised manuscript.

Kind regards,

Morteza Arab-Zozani, Ph. D.

Academic Editor

PLOS ONE

Reviewers' comments:

Reviewer's Responses to Questions

**Comments to the Author**

1. Does the manuscript provide a valid rationale for the proposed study, with clearly identified and justified research questions?

Reviewer #1: Yes

Reviewer #2: Yes

Reviewer #3: Yes

2. Is the protocol technically sound and planned in a manner that will lead to a meaningful outcome and allow testing the stated hypotheses?

Reviewer #1: Yes

Reviewer #2: Yes

Reviewer #3: Yes

3. Is the methodology feasible and described in sufficient detail to allow the work to be replicable?

Reviewer #1: Yes

Reviewer #2: Yes

Reviewer #3: Yes

4. Have the authors described where all data underlying the findings will be made available when the study is complete?

Reviewer #1: Yes

Reviewer #2: Yes

Reviewer #3: Yes

5. Is the manuscript presented in an intelligible fashion and written in standard English?

Reviewer #1: Yes

Reviewer #2: Yes

Reviewer #3: Yes

6. Review Comments to the Author

You may also provide optional suggestions and comments to authors that they might find helpful in planning their study.

Reviewer #1: This study will contribute to the topic of positive youth development via offering guidance regarding the optimal scales for developmental assets. And the protocol is well-organized and can be accepted after revising the citation format in the text.

Reviewer #2: Thanks for receiving the opportunity to review your paper. Overall, I appreciate the thoroughness and importance of your study in assessing the psychometric properties of developmental assets scales. Your focus on systematically reviewing and conducting a meta-analysis adds significant value to the existing literature. However, the manuscript must undergo minor revisions suggested below before it can be considered for publication.

I suggest a minor adjustment in the structure of the Abstract. I recommend you to consider incorporating your key results and conclusions section in place of the discussion section.

The introduction provides comprehensive background information that is highly relevant to the study. However, the authors must significantly expand on it by highlighting the significance and novelty of the study.

In methodology, address how you will assess and manage heterogeneity among the included studies. Mention all the statistical tests used in the study.

I noted an issue with the PRISMA flowchart presented as Figure [1]. It appears that the chart is incomplete as the value of 'k' (number of studies included in the meta-analysis) is not reflecting accurately. This discrepancy undermines the clarity and transparency of your methodology, which are essential aspects of systematic reviews and meta-analyses. I kindly request that you review and revise the PRISMA flowchart to ensure that it accurately represents the number of studies included in your analysis. Providing an accurate depiction of the study selection process is crucial for readers to assess the rigor and comprehensiveness of your review.

In result section, the data are well-organized. However, I suggest to add a well-constructed forest plot which can help to concisely convey the results of a meta-analysis. Incorporating this graphical depiction will not only aid in understanding the individual study contributions but also provide a holistic view of the overall effect size and its variability. Ensure consistency in the representation of odds ratios across the forest plot.

In discussion section, it should be more objective and clearer according to the purpose of the study and focus more on potential implications of your study's findings for both research and practice. Also add future research direction based on current findings.

Expand the reference list as systematic reviews and meta-analyses are expected to include a comprehensive collection of literature.

I suggest English language revisions and other typos and grammar errors should be corrected.

Reviewer #3: The authors performed a study addressing “Systematic Review and Meta-Analysis of Developmental Assets Scales: A Study Protocol for Psychometric Properties”. I didn't find any major problems with your manuscript.

7. PLOS authors have the option to publish the peer review history of their article (what does this mean?). If published, this will include your full peer review and any attached files.

Reviewer #1: No

Reviewer #2: No

Reviewer #3: No

---

## [Author Response · Author response to Decision Letter 0]

7 Apr 2024

Reviewer #2: 

Thanks for receiving the opportunity to review your paper. Overall, I appreciate the thoroughness and importance of your study in assessing the psychometric properties of developmental assets scales. Your focus on systematically reviewing and conducting a meta-analysis adds significant value to the existing literature. However, the manuscript must undergo minor revisions suggested below before it can be considered for publication.

1. I suggest a minor adjustment in the structure of the Abstract. I recommend you to consider incorporating your key results and conclusions section in place of the discussion section.

Response: We appreciate your insightful suggestions regarding the structure of the abstract. However, we would like to clarify that this study is a protocol study, and as such, we have not yet generated any results to include in the abstract. Nevertheless, we will ensure that the abstract effectively outlines the aims, methods, and anticipated contributions of our protocol study.

2. The introduction provides comprehensive background information that is highly relevant to the study. However, the authors must significantly expand on it by highlighting the significance and novelty of the study.

Response: Thank you for your attention to detail. We have addressed the significance and novelty of our study on Lines 118-141 and Lines 376-379 of the manuscript.

3. In methodology, address how you will assess and manage heterogeneity among the included studies. Mention all the statistical tests used in the study.

Response: Thank you for requesting this clarification. We will address heterogeneity among included studies through subgroup analysis, sensitivity analysis, and potentially meta-regression. Statistical tests will include meta-analyses for pooled effect sizes and assessments for publication bias. We appreciate your concise guidance and will ensure these aspects are clearly outlined in the methodology section of our manuscript. This has been clarified on lines 273-281.

4. I noted an issue with the PRISMA flowchart presented as Figure [1]. It appears that the chart is incomplete as the value of 'k' (number of studies included in the meta-analysis) is not reflecting accurately. This discrepancy undermines the clarity and transparency of your methodology, which are essential aspects of systematic reviews and meta-analyses. I kindly request that you review and revise the PRISMA flowchart to ensure that it accurately represents the number of studies included in your analysis. Providing an accurate depiction of the study selection process is crucial for readers to assess the rigor and comprehensiveness of your review.

Response: We appreciate your suggestions regarding the PRISMA flowchart. However, we would like to clarify that this study is a protocol study which involves no study selection process. Therefore, as a standard practice for study protocols and a requirement of the PLOS ONE, an incomplete PRISMA flowchart is presented.

5. In result section, the data are well-organized. However, I suggest to add a well-constructed forest plot which can help to concisely convey the results of a meta-analysis. Incorporating this graphical depiction will not only aid in understanding the individual study contributions but also provide a holistic view of the overall effect size and its variability. Ensure consistency in the representation of odds ratios across the forest plot.

Response: We appreciate your suggestions. However, as explained above, we would like to clarify that this study is a protocol study, and as such, we generated no results and forest plots yet.

5. In discussion section, it should be more objective and clearer according to the purpose of the study and focus more on potential implications of your study's findings for both research and practice. Also add future research direction based on current findings.

Response: We appreciate this comment. The study objectives are outlined in the discussion part on lines 345-352. As for the potential implications for research and practice, we have explained them in the discussion part on lines 365-387.

6. Expand the reference list as systematic reviews and meta-analyses are expected to include a comprehensive collection of literature.

Response: As the studies on psychometric properties of the developmental assets scales were limited in numbers, and as this study protocol has not conducted a systematic review of all such studies (that gives us a list of them), not all of these studies are cited in the present protocol. However, the main and primary studies of the scales [e.g., Benson, Scales, and Syvertsen (2011); Oman et al. (2002; 2018)], as well as the theoretical studies in this field [e.g., Catalano et al. (2004); Benson, Leffert, Scales, and Blyth (1998)] are included in the introduction and discussion parts. 

7. I suggest English language revisions and other typos and grammar errors should be corrected.

Response: We appreciate the thorough review and have reexamined the entire manuscript for spelling errors. Moreover, one of our coauthors, a US native scholar with expertise in PYD and developmental psychopathology (ETC), has re-reviewed the manuscript for readability.

Note:

Response: We have now changed the format of Fig 1 to “tif” using the PACE and have uploaded it beside the revised manuscript.

---

## [Editor Report · Decision Letter 1]

21 Aug 2024

Systematic Review and Meta-Analysis of Developmental Assets Scales: A Study Protocol for Psychometric Properties

PONE-D-23-40747R1

Dear Dr. Mojtaba Habibi Asgarabad,

We’re pleased to inform you that your manuscript has been judged scientifically suitable for publication and will be formally accepted for publication once it meets all outstanding technical requirements.

Kind regards,

Chrysanthi Lioupi

Guest Editor

PLOS ONE

---

## [Editor Report · Acceptance letter]

30 Aug 2024

PONE-D-23-40747R1 

PLOS ONE

Dear Dr. Habibi Asgarabad, 

I'm pleased to inform you that your manuscript has been deemed suitable for publication in PLOS ONE. Congratulations! Your manuscript is now being handed over to our production team.

Kind regards, 

on behalf of

Dr. Chrysanthi Lioupi 

Guest Editor

PLOS ONE